# School-Level Variation in Coverage of Co-Administered dTpa and HPV Dose 1 in Three Australian States

**DOI:** 10.3390/vaccines9101202

**Published:** 2021-10-19

**Authors:** Cassandra Vujovich-Dunn, Susan Rachel Skinner, Julia Brotherton, Handan Wand, Jana Sisnowski, Rebecca Lorch, Mark Veitch, Vicky Sheppeard, Paul Effler, Heather Gidding, Alison Venn, Cristyn Davies, Jane Hocking, Lisa J. Whop, Julie Leask, Karen Canfell, Lena Sanci, Megan Smith, Melissa Kang, Meredith Temple-Smith, Michael Kidd, Sharyn Burns, Linda Selvey, Dennis Meijer, Sonya Ennis, Chloe A. Thomson, Nikole Lane, John Kaldor, Rebecca Guy

**Affiliations:** 1The Kirby Institute, University of New South Wales, Kensington, Sydney 2052, Australia; Hwand@kirby.unsw.edu.au (H.W.); jsisnowski@kirby.unsw.edu.au (J.S.); rlorch@kirby.unsw.edu.au (R.L.); jkaldor@kirby.unsw.edu.au (J.K.); Rguy@kirby.unsw.edu.au (R.G.); 2Children’s Hospital Westmead, Sydney Children’s Hospitals Network, Sydney 2145, Australia; rachel.skinner@sydney.edu.au (S.R.S.); cristyn.davies@sydney.edu.au (C.D.); 3Faculty of Medicine and Health, University of Sydney, Specialty of Child and Adolescent Health, Sydney 2006, Australia; 4Population Health, VCS Foundation Ltd., East Melbourne, Melbourne 3053, Australia; jbrother@vcs.org.au; 5Melbourne School of Population and Global Health, University of Melbourne, Carlton, Melbourne 3010, Australia; j.hocking@unimelb.edu.au; 6National Centre for Epidemiology & Population Health, Australian National University, Canberra 0200, Australia; lisa.whop@anu.edu.au; 7Department of Health and Human Services, Tasmanian Government, Hobart 7001, Australia; mark.veitch@health.tas.gov.au (M.V.); nikki.lane@health.tas.gov.au (N.L.); 8Communicable Diseases Branch, Health Protection NSW, St Leonards, Sydney 2065, Australia; Vicky.Sheppeard@health.nsw.gov.au; 9School of Public Health, University of Sydney, Camperdown, Sydney 2006, Australia; Megan.Smith@nswcc.org.au; 10Department of Health, Communicable Disease Control Directorate, East Perth 6000, Australia; Paul.Effler@health.wa.gov.au (P.E.); Chloe.Thomson@health.wa.gov.au (C.A.T.); 11School of Population Health, University of New Souh Wales, Sydney 2052, Australia; heather.gidding@sydney.edu.au; 12Norther Clinical School of Sydney, University of Sydney, Camperdown, Sydney 2006, Australia; 13Women and Babies Research, Kollin Intstitye, Northern Sydney Local Health District, St Leaonards, Sydney 2064, Australia; 14National Centre for Immunisation Research and Surveillance, Westmead, Sydney 2145, Australia; 15Menzies Institute for Medical Research, University of Tasmania, Tasmanian 7000, Australia; alison.venn@utas.edu.au; 16Menzies School of Health Research, Charles Darwin University, Brisbane 4000, Australia; 17Faculty of Medicine and Health, Sydney Nursing School, University of Sydney, Camperdown, Sydney 2006, Australia; julie.leask@sydney.edu.au; 18Cancer Research Division, Cancer Council, Brisbane 2011, Australia; karen.canfell@nswcc.org.au; 19Medicine, Dentistry and Health Sciences, University of Melbourne, Carlton, Melbourne 3010, Australia; l.sanci@unimelb.edu.au (L.S.); m.temple-smith@unimelb.edu.au (M.T.-S.); 20Westmead Clinical School, University of Sydney, Sydney 2006, Australia; Melissa.Kang@uts.edu.au; 21Southgate Institute for Health, Flinders University, Adelaide 5042, Australia; michael.kidd@flinders.edu.au; 22School of Population Health, Curtin University, Perth 6102, Australia; s.burns@curtin.edu.au; 23School of Public Health, University of Queensland, Brisbane 4072, Australia; l.selvey@uq.edu.au; 24Immunisation Unit, Health Protection NSW, St Leonards, Sydney 2065, Australia; Dennis.Meijer@health.nsw.gov.au (D.M.); Sonya.Ennis@health.nsw.gov.au (S.E.)

**Keywords:** vaccination, implementation, school-based immunisation, evaluation and impact, vaccine specific hesitancy, differential uptake, adolescent vaccination, cancer prevention

## Abstract

Background: Australian adolescents are routinely offered HPV and dTpa (diphtheria, tetanus, pertussis) vaccines simultaneously in the secondary school vaccination program. We identified schools where HPV initiation was lower than dTpa coverage and associated school-level factors across three states. Methods: HPV vaccination initiation rates and dTpa vaccination coverage in 2016 were calculated using vaccine databases and school enrolment data. A multivariate analysis assessed sociodemographic and school-level factors associated with HPV initiation being >5% absolute lower than dTpa coverage. Results: Of 1280 schools included, the median school-level HPV initiation rate was 85% (interquartile range (IQR):75–90%) and the median dTpa coverage was 86% (IQR:75–92%). Nearly a quarter (24%) of all schools had HPV vaccination initiation >5% lower than dTpa coverage and 11 % had >10% difference. School-level factors independently associated with >5% difference were remote schools (aOR:3.5, 95% CI = 1.7–7.2) and schools in major cities (aOR:1.8, 95% CI = 1.0–3.0), small schools (aOR:3.3, 95% CI = 2.3–5.7), higher socioeconomic advantage (aOR:1.7, 95% CI = 1.1–2.6), and lower proportions of Language-background-other-than-English (aOR:1.9, 95% CI = 1.2–3.0). Conclusion: The results identified a quarter of schools had lower HPV than dTpa initiation coverage, which may indicate HPV vaccine hesitancy, and the difference was more likely in socioeconomically advantaged schools. As hesitancy is context specific, it is important to understand the potential drivers of hesitancy and future research needs to understand the reasons driving differential uptake.

## 1. Introduction

Human papillomavirus (HPV) is the most common sexually transmissible infection (STI) globally and is associated with significant morbidity and mortality due to cervical and other cancers and genital warts, with 50–80% of the general population acquiring HPV infection with one or more types at some point in their life [1]. Infection rates are highest in younger females in their early to mid-20s following sexual debut [2]. Most HPV infection is asymptomatic and resolves within 1–2 years. However, persistent infection with an oncogenic type of HPV can sometimes, usually over decades, lead to the development of cervical cancer, as well as anal, vulvar, vaginal, penile and oropharyngeal cancer [3,4]. Cervical cancer is the fourth most common cause of cancer incidence and mortality in women worldwide [5]. In addition, around 90% of all cases of genital warts are caused by the non-oncogenic HPV types 6 and 11 [4] and prior to vaccination, genital warts were the most commonly managed STI at sexual health clinics contributing to major psychosocial impacts and costs [6,7,8]. 

Three vaccines that protect against oncogenic types of HPV have been, or are currently, licensed in most countries: a bivalent vaccine; a quadrivalent vaccine [9], and a nonavalent vaccine [10], with the quadrivalent and nonavalent vaccines also providing protection against genital warts. The primary aim of HPV vaccination is to prevent cervical cancer [11]. In 2018, the World Health Organization (WHO) called for countries to work towards the global elimination of cervical cancer as a public health problem [12,13] given the effectiveness of the vaccines [14,15,16], and cervical screening using HPV testing. Furthermore, the WHO Global Health Sector Strategy on STIs highlights the importance of HPV vaccination to achieve the elimination of genital warts [17]. In most countries, HPV immunisation programs are targeted at young adolescents, as the vaccine is most effective prior to initiation of sexual activity and exposure to the virus [18,19,20]. Voluntary on-site school-based HPV vaccination programs have been successful in achieving high rates of vaccination coverage in adolescents in Australia, Canada, several European nations, Malaysia, Brunei and other low middle-income countries [21,22,23,24,25,26].

Australia was the first country to implement a national school-based HPV vaccination program. Since its introduction, it has achieved high coverage of HPV vaccination, reaching just over 80% of 15-year-olds in 2017 for the three-dose course [27]. Vaccines are offered in schools at vaccine clinics, following parental/guardian consent. The HPV vaccine can also be accessed in general practice (predominantly) or other primary health care services such as public sector Community Health Centres but are not routinely offered in this setting due to the free school-based immunisation program. Despite the high HPV coverage achieved in Australia through the school program, there is variation in coverage between jurisdictions and also in smaller geographical areas within jurisdictions [28,29]. The lower coverage in some areas or populations may signal hesitancy towards the HPV vaccination. Vaccine hesitancy is a motivational state of being conflicted about or opposed to getting vaccinated [30]. It may be associated with safety concerns or lack of familiarity with a vaccine [31,32]. Differences in HPV vaccination coverage compared to other adolescent vaccinations may signal specific concerns around the HPV vaccine for parents and adolescents. For example, parents in some countries have objected to HPV vaccination on the grounds that it is offered to protect against an STI and may lead to sexual disinhibition, although such disinhibition has not been demonstrated in the literature [33]. The WHO lists vaccine hesitancy as one of the ten threats to global health in 2019 [34], yet impact of hesitancy on HPV vaccination uptake in Australia is not well understood [35].

One objective marker of hesitancy towards the HPV vaccination specifically (rather than vaccines in general) is if parents refuse the HPV vaccine but consent to other adolescent vaccinations offered at the same time. In Australia we have a unique opportunity to measure this, as HPV vaccine is co-administered in schools with the diphtheria tetanus pertussis acellular (dTpa) vaccine. The dTpa vaccine was implemented as a booster for adolescents in Australia in 2004, building on a long-standing childhood vaccination program underway since 1986 [36]. HPV vaccination was added as an adolescent vaccine in 2007 for girls and 2013 for boys [10].

A previous ecological study from the United Kingdom, found differential coverage between the HPV vaccine and the meningococcal A, C, W and Y (MenACWY) vaccine [37]. However, this study looked at difference in vaccine coverage which included both doses of the HPV vaccine, and potentially doses of HPV and MenACWY from the previous year. To our knowledge, our study is the first to compare initiation coverage after a co-consent process of adolescent vaccines, where HPV vaccine is delivered through the school-based immunisation program. This is an important distinction as our study provides insight to the decision-making process when co-consenting and co-administering vaccines and may be a marker of vaccine specific hesitancy.

In this study, we aimed to estimate the school-level coverage of dose one of the HPV vaccine and the dTpa booster, in three Australian states, and identify school-level correlates associated with lower relative coverage of the HPV vaccine. 

## 2. Materials and Methods

### 2.1. Study Design and Context

We conducted an ecological analysis of HPV and dTpa vaccine coverage across three Australian states—New South Wales (NSW), Tasmania (TAS) and Western Australia (WA), with the school as the unit of analysis. 

### 2.2. Study Population

We included first year secondary school students aged 11–13 years enrolled in all secondary schools that provide education to adolescents from around 11 to 18 years of age in the three states, and only included those in the first year of secondary school, generally aged 11–13 years, and schools for which both year-specific student enrolment numbers, as well as delivered HPV vaccination doses and dTpa doses were available for 2016. 

### 2.3. Study Outcome

The primary outcome for each school was binary, defined by whether the proportion of adolescents in the school who received the first dose of HPV vaccine (i.e., initiation of the HPV vaccination course, designated as ‘HPV coverage’ hereafter) in 2016 was more than 5% lower than the proportion who received dTpa in the same year (dTpa coverage), hereafter designated as ‘lower HPV coverage than dTpa’. The denominator for each proportion was the total school enrolments. We chose a difference of 5% or more between HPV and dTpa coverage as it was considered clinically meaningful given the high overall coverage of HPV vaccination in Australia.

### 2.4. Data Sources

We obtained data from the following four existing administrative datasets held by state and national statutory bodies to calculate the study outcomes and co-variates. All datasets were at the school or postcode level.

1. The National HPV Vaccination Program Register (NHPVR): until 2018, collected details of individual doses of HPV vaccine delivered in Australian schools and general practice/primary health care services as notified to the register as part of the National Immunisation Program (NIP). We obtained vaccination doses given by cohort year and school name. The register did not collect information on dTpa vaccines.

2. Jurisdictional health departments: collect individual data on HPV and dTpa doses delivered in schools. We obtained dTpa vaccination doses and school enrolment data for each school grade in which the vaccine program was delivered (Year 7 in two jurisdictions, Year 8 in one jurisdiction). HPV doses were not requested, as these were available through the NHPVR. Enrolment data collected from NSW, WA and TAS included graded schools (which denoted school year) for mainstream schools and special education schools. WA also included ungraded ‘special education’ schools (schools that do not place students in a specific year level).

3. The Australian Curriculum, Assessment and Reporting Authority (ACARA): collects and reports characteristics of schools and their students [38]. 

4. The Australian Bureau of Statistics: is a national statutory body which collects demographic data from Australian residents by a Census every five years [39], from which we obtained SEIFA Index of relative Socio-Economic Disadvantage. 

### 2.5. Statistical Analysis

We first merged the datasets above deterministically based on the school name and postcode, and then for those remaining unmatched via probabilistic matching using the STATA reclink command [30]. Remaining records were reviewed manually, and plausible matches were included. 

We then conducted descriptive analyses of school-level HPV coverage and dTpa coverage at the school-level for the three jurisdictions. This included calculation of the overall coverage of both vaccines for each school. Across schools this included calculation of means, standard deviations, medians, ranges, and interquartile ranges (IQR). We calculated the proportion of schools where HPV vaccine was lower than dTpa, and the proportion of schools where dTpa was lower than HPV vaccine, and for each categorised the differences as +/− 0%, 0–5%, >5–10% and >10% difference.

We conducted univariate and multivariate analysis to identify sociodemographic and school-related factors associated with the primary outcome of ‘lower HPV coverage’ defined as HPV coverage that was at least five percentage points lower than dTpa coverage. Sociodemographic variables included the percent of enrolments with Aboriginal and/or Torres Strait Islander background (referred to hereafter as Indigenous students), the percent of enrolments with a language-background-other-than-English, and the Index of Relative Socio-Economic Disadvantage (SEIFA IRSD) (a standard area level measure of disadvantage summarising a range of information about economic and social conditions of households within an area), defined by postcode of the school [39]. School-related factors included co-educational status (school enrolments of both males and females, or single sex schools), special education status (special educational schools or mainstream schools), geographical location of the school (major city, inner regional, outer regional, remote, and very remote), size of the school (based on total enrolments) and attendance (% of all possible school days attended). We categorised the continuous measurements (school size, Indigenous student enrolments, language-background-other-than-English, socioeconomic disadvantage score and attendance) using tertiles (Table 1).

Variables associated with the primary outcome were first assessed using univariate logistic regression models. The reference categories were selected based on normative groups that have been identified in the literature as not having a tendency towards vaccination hesitancy (e.g., previous studies have identified that parents of higher socioeconomic status are more likely to be hesitant, so low socioeconomic status was chosen as the reference category) [40,41,42]. Based on this literature, we selected reference categories of inner regional for remoteness, low socioeconomic status postcodes, and higher proportion of English as a second language background, and larger schools. For the Indigenous student enrolment co-variate, the middle tertile category was selected as reference category because it most closely represents the general population distribution of Indigenous Australians, which was estimated as 3.3% in 2016 [43]. Finally, variables without a normative group category were selected based on a large enough sample in the group (co-education status and mainstream schools).

Variables with a *p*-value < 0.10 were included in the initial full multivariable model. Covariates with overall statistical significance of *p* < 0.05 based on the test for heterogeneity were included in the final reduced models. We present the odds ratios and 95% CIs. We restricted the analysis to schools with a vaccine-eligible enrolment of at least 10 students to minimise the influence of schools with very small numbers. The final multivariable model was also adjusted for a five-level school affiliation variable, which included Independent, Catholic, Government schools and jurisdiction. However, to maintain anonymity as requested by stakeholders in the education sector, coverage and correlation by school affiliation is not reported publicly but the results were provided to relevant program managers to inform future policy priorities. 

### 2.6. Ethical Approval

Ethical approval was provided by the Human Research Ethics Committees of the University of New South Wales (HC17632), the Australian National University (2017/516), the University of Tasmania (1320/17), the Aboriginal Health and Medical Research Council of New South Wales (1320/17), the Aboriginal Health Council of Western Australia (818), and the Department of Health of Western Australia (RGS0000000456).

## 3. Results

Of 1327 schools in the dataset, we excluded 39 schools with no 2016 enrolment data, two not located in a participating jurisdiction, and the remainder because the school name was not available in both the numerator and denominator data to enable matching of the datasets with the NHPVR. An additional six schools were excluded as they did not report on doses of dTpa given, leaving 1280 schools included in the final analysis.

Of the schools included, 58% were in major cities, 86% were co-educational, 8% were special education schools, 31% had a high proportion of Indigenous student enrolments (9–100%), 33% were small schools (enrolment of 11–383 students), 33% were located in areas in the most socioeconomically disadvantaged tertile and 35% had low attendance rates (29–87% of all possible school days attended) (see Table 1). The median school-level HPV coverage was 85% (interquartile range (IQR: 75–90%) and the median dTpa coverage was 86% (IQR: 75–92%). Of 1280 schools included, 200 schools (16%) had no difference in vaccination uptake between HPV and dTpa; 53% (*n* = 674) had lower HPV than dTpa coverage and 32% (*n* = 410) lower dTpa than HPV coverage (Figure 1). Of the 674 schools where HPV coverage was lower than dTpa coverage, 295 had a difference of more than 5% (‘lower HPV coverage’ group), and of these schools, 93 (32%) had HPV vaccine coverage less than 70% (Figure 2). 

In the univariate analysis, sociodemographic factors associated with lower HPV coverage were: schools with a lower proportion of Indigenous student enrolments (OR 1.53, 95% CI = 1.1–2.2), schools with a lower proportion of adolescents from non-English language backgrounds (OR 2.2, 95% CI = 1.6–3.1) and schools with a higher proportion of adolescents from areas of greatest socioeconomic advantage (OR 2.0 95% CI = 1.4–2.8) (Table 2). School factors associated with lower HPV coverage were: located in remote/very remote areas (OR 2.9, 95% CI = 1.6–5.3) or major cities (OR 2.2, 95% CI = 1.5–3.2), and small (OR 1.9, 95% CI = 1.4–2.7) or medium enrolment size (OR 1.0, 95% CI = 0.7–1.5). Collinearity was detected between school size, remoteness, and socioeconomic disadvantage, and between lower attendance and Indigenous student enrolments. The final multivariate model therefore included remoteness, school size, Indigenous student enrolment, non-English language background, and socioeconomic advantage. 

In the multivariate analysis, sociodemographic factors independently associated with lower HPV coverage were: schools with a lower proportion of Indigenous student enrolments (aOR 0.6, 95% CI = 0.4–1.0), a lower proportion of adolescents from non-English-language-backgrounds (aOR 1.9, 95% CI = 1.2–3.0), and a higher proportion of adolescents from areas of greatest socioeconomic advantage (aOR 1.7, 95% CI = 1.1–2.6). School related factors independently associated lower HPV coverage were: location in remote areas (aOR 3.5, 95% CI = 1.7–7.2), or major cities (aOR 1.8, 95% CI = 1.0–3.0); and small school size (aOR 3.3, 95% CI = 2.3–5.7) or medium size (aOR 1.5, 95% CI = 1.0–2.3) (Table 2). 

## 4. Discussion

We identified variation in the uptake of HPV and dTpa vaccines in the school-based immunisation program, with HPV coverage at least 5% lower than dTpa coverage in 23% of schools in the three states. To our knowledge this is the first study to compare coverage of these two school-based vaccines in Australia and examine school-level factors associated with differences in their coverage. We found that schools with a higher proportion of adolescents from areas with higher socio-economic advantage and from English-speaking backgrounds were more likely to have lower HPV coverage than dTpa coverage. We also found that the size of the school was an independent predictor, with small- and medium-sized schools more likely to report a greater than 5% difference between HPV dose 1 and dTpa than larger schools. 

More than 14% of schools had a substantially lower HPV coverage of more than 10%, suggesting the difference is an important contributor to lower uptake at the population level. We also found that in 32% of schools dTpa coverage was lower than HPV, which may potentially be explained by adolescents having received a diphtheria tetanus vaccine previously for tetanus prevention, if a child or adolescent had a recent cut or wound that could have been in contact with soil, or routine travel vaccination. In Australia, children aged <10 years with a tetanus-prone wound are recommended to receive DTPa or a DTPa combination vaccine, and people aged ≥10 years are recommended to receive a booster dose of dT or dTpa if their last dose was more than 5 years ago [44].

Schools in our study were more likely to report a larger difference between dTpa and HPV if they were small or medium sized schools and had enrolment between 11 and 844, in comparison to larger schools in the study. A study from the UK that found differential coverage between the HPV vaccine and the meningococcal A, C, W and Y (MenACWY) vaccine, also identified smalls schools as achieving lower coverage [37]. It is currently not known, why small schools have lower coverage and higher differential uptake compared to other schools. However, it may be that smalls schools have multiple risk factors co-occurring relating to collinearity. In our study, most special schools were small, and most small schools were also located in remote areas. 

We found that several sociodemographic characteristics relating to the school were strongly associated with lower HPV coverage than dTpa coverage, including being located in less disadvantaged areas and a higher proportion of students from English-speaking backgrounds. Studies from the US and Denmark have also found that parents and girls from higher socio-economic areas were more likely to be hesitant about HPV vaccination [45,46]. The Danish study found girls living in municipalities with higher disposable income were more likely to have fluctuating responses, and eventual negative responses to negative media coverage of the HPV vaccination in Denmark [46]. However, a recent study of adolescent vaccines delivered in the school-based program in the United Kingdom found that schools located in the least deprived areas had highest coverage for HPV and MenACWY [37]. In Australia, a recent study identified that lower vaccination uptake in children under five was associated with postcodes in areas with greater advantage, and among parents with greater education and occupational status [47]. In the Australian study, the authors suggested these parents were more likely to be exposed to anti-vaccination attitudes via social networks. However, this does not explain the difference in uptake between the two vaccines. Previous research has reported mixed results on associations between vaccination uptake and ethnicity or immigration status [48,49,50,51,52] but none have specifically focused on differences between HPV and another vaccine delivered at around the same time. Our study did not assess vaccination uptake and religiosity, however the ecological study from the UK assessing vaccine coverage of HPV and MenACWY found that Jewish and Muslims schools had lower coverage of HPV but not MenACWY, suggesting there were not issues in this schools with vaccination acceptance in general, but may be lower vaccine acceptance for HPV specifically [37]. This trend was not seen in the Roman Catholic schools or non-denominational schools where there was no reported differential uptake [37].

We also found that schools with a very small proportion of Indigenous students enrolled were more likely to have HPV coverage that was lower than dTpa (although the latter was not statistically significant). In a recent analysis we conducted from this same dataset, we found schools with a higher proportion of Indigenous student enrolments were associated with a lower absolute uptake of HPV, both initiation and completion [53]. Schools with a higher percent of Indigenous student enrolment were also schools that were most likely to be categorized as having low attendance. It is reasonable to assume if students are absent on the day that vaccines are administered, they would miss both vaccines, which may explain why there was less difference between HPV dose 1 and dTpa booster in schools with the highest Indigenous student enrolment. The reason for lower initiation in these schools may have more to do with process and opportunity barriers rather than vaccine hesitancy or related issues, as noted in the literature [54,55,56,57,58,59,60].

We identified some school related factors associated with lower HPV coverage than dTpa in our analysis. School size has previously been reported as independently associated with differential uptake of adolescent vaccines, as well as lowest coverage of vaccines compared to larger schools [37]. The association with small school size may have been related to the increased likelihood of detecting a difference of >5% when dealing with small numbers. However, our analysis excluded schools with <10 students and the association remained. The reason for the increased likelihood of the >5% difference being detected in schools located in major cities and remote locations, is unknown, but may be a marker of other unmeasured factors. 

Our study has a few limitations to consider when interpreting the findings. First, we used an ecological approach to identify schools and specific characteristics associated with relatively lower uptake of HPV compared to dTpa. Therefore, our approach does not assess risk factors at the individual level, but rather provides an indication of school-level factors which can inform interventions and future research. The co-variates were based on those routinely available in a range of datasets, and some may have been markers of other risk factors. For example, we measured English language backgrounds rather than more nuanced factors such as ethnicity or cultural or religious background. Also, some variables were only available at the postcode rather than the school level, and in some postcodes, there is considerable variation in the sociodemographic of the population. Third, many of the variables describing school characteristics were highly correlated, which may have led to an inability to resolve confounding and distinguish causally related factors. Fourth, the HPV and dTpa uptake calculation only included vaccinations that could be attributed to a school, which was necessary for the analysis, but may have somewhat under-estimated true vaccine coverage. Lastly, coverage of HPV and dTpa was not obtained from the same data source, but our calculations of HPV coverage from the National HPV Vaccination Program Register is similar to those reported by the jurisdictions separately [61,62], so we do not believe this would have introduced measurement bias.

## 5. Conclusions

In conclusion, our study identified several key sociodemographic characteristics and school related factors that were associated with HPV coverage that is >5% lower than another adolescent vaccine co-administered in the school setting. This is an important finding as it may indicate a degree of hesitancy specifically to the HPV vaccine in some contexts within Australia, which has previously not been identified in the literature. It is important to conduct further research to confirm if hesitancy is contributing to lower HPV vaccine coverage in some schools, and the potential drivers of hesitancy [35]. The results provide a novel perspective to the existing gaps in HPV vaccination. Research needs to be prioritised to understand the reasons driving differential uptake, and to explore potential causal pathways contributing to associations between school characteristics and >5% difference between HPV dose 1 and dTpa.

## Figures and Tables

**Figure 1 vaccines-09-01202-f001:**
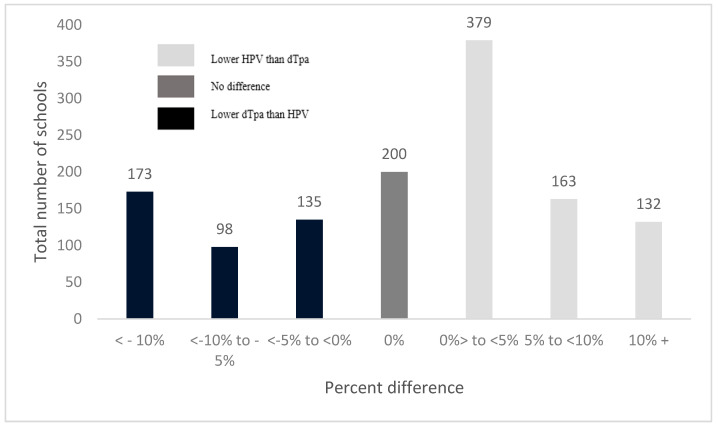
Percent difference between dTpa and HPV coverage at the school level, 2016 (*n* = 1280).

**Figure 2 vaccines-09-01202-f002:**
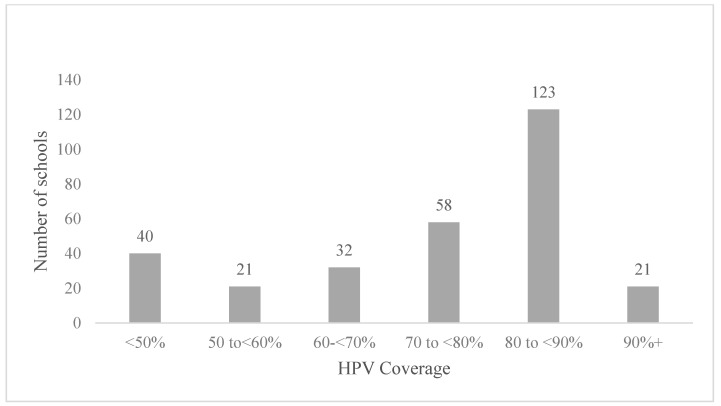
HPV coverage at schools where HPV coverage was lower^1^ than dTpa coverage, 2016 (*n* = 295). ^1^ HPV coverage that is more than 5% lower than dTpa coverage.

**Table 1 vaccines-09-01202-t001:** Characteristics of schools included in the analysis, 2016 ^1^.

Variable	*n*	(%)
**Remoteness classification**		
Major cities	750	58
Inner Regional	288	22
Outer Regional	165	13
Remote and very remote	72	6
Missing	5	0.4
**Co-educational status**		
Co-educational school	1100	86
Single sex school	150	12
Missing	30	2
**% Indigenous student enrolment ^2^**		
Low (0–2%)	451	35
Medium (3–8%)	341	27
High (9–100%)	397	31
Missing	97	8
**Relative socioeconomic disadvantage score** **(postcode)**		
Most disadvantaged	423	33
Less disadvantaged	428	33
Least disadvantaged	423	33
Missing	6	0.5
**School size (total school enrolment) ^2^**		
Small (11–383)	415	32
Medium (384–844)	419	33
Large (845–2735)	416	33
Missing	30	2
**Special education status**		
Mainstream school	1172	92
Special education school	103	8
Missing	5	0.4
**Language-background-other-than-English ^2^**		
Low (0–6%)	451	35
Medium (7–22%)	380	30
High (23–100%)	413	32
Missing	36	3
**Attendance (% of all possible school days** **attended) ^2^**		
Low (29–87%)	449	35
Medium (88–91%)	417	33
High (92–97%)	272	21
Missing	142	11

^1^ Proportions were calculated for non-missing data only. ^2^ Continuous measurements were categorised using tertiles.

**Table 2 vaccines-09-01202-t002:** Correlates of >5% lower HPV coverage than dTpa at the school level, 2016 (*n* = 1280).

Variable	HPV Coverage > 5% Lower than dTpa N (%)	Univariate Analysis		Final Multivariate Model ^3^	
		OR (95% CI)	*p*-Value ^1^	aOR (95% CI) ^2^	*p*-Value ^1^
**Remoteness area**					
Major cities	196 (26%)	2.2 (1.5–3.2)	0.000	**1.8 (1.0–3.0)**	**0.036**
Inner Regional	40 (14%)	Ref ^4^		Ref	
Outer Regional	35 (20%)	1.6 (0.9–2.6)	0.009	**2.3 (1.3–4.3)**	**0.007**
Remote and very remote	23 (32%)	2.9 (1.6–5.3)	0.000	**3.5 (1.7–7.2)**	**0.007**
**School size (total school enrolment) ^5^**					
Small (11–383)	125 (30%)	1.9 (1.4–2.7)	0.000	**3.3 (2.3–5.7)**	<0.001
Medium (384–844)	79 (19%)	1.0 (0.7–1.5)	0.828	**1.5 (1.0–2.3)**	**0.034**
Large (845–2735)	76 (18%)	Ref		Ref	
**Co-ed status**					
Co-ed school	247 (22%)	Ref		Ref	
Single sex school	33 (22%)	1.0 (0.7–1.5)	0.900	0.7 (0.5–1.2)	0.235
**Special education status**					
Mainstream school	262 (22%)	Ref		Ref	
Special education school	30 (29%)	1.4 (0.9–2.2)	0.118	1.2 (0.6–2.2)	0.623
**% Indigenous student enrolment ^5^**					
Low (0–2%)	123 (27%)	1.5 (1.1–2.2)	0.013	1.4 (1.0–2.1)	0.091
Medium (3–8%)	67 (20%)	Ref		Ref	
High (9–100%)	61 (16%)	0.8 (0.5–1.1)	0.151	**0.6 (0.4–1.0)**	**0.040**
**% non-English language background ^5^**					
Low (0–6%)	68 (15%)	1.9 (1.3–2.6)	0.001	**1.6 (1.1–2.5)**	**0.030**
Medium (7–22%)	94 (25%)	2.2 (1.6–3.1)	0.000	**1.9 (1.2–3.0)**	**0.011**
High (23–100%)	117 (28%)	Ref		Ref	
**School postcode score, Relative Socioeconomic Disadvantage**					
Low (604–967)	72 (16%)	Ref		Ref	
Medium (698–1016)	99 (23%)	1.5 (1.1–2.1)	0.027	1.4 (1.0–2.2)	0.091
High ^6^ (1017–1128)	122 (29%)	2.0 (1.4–2.8)	0.000	**1.7 (1.1–2.6)**	**0.023**
**Attendance rate (% of all possible school days attended) ^5^**					
Low (29–87%)	72 (16%)	Ref		Ref	
Medium (88–91%)	101 (24%)	1.7 (1.2–2.3)	0.003	1.3 (0.8–2.0)	0.257
High (92–97%)	71 (26%)	1.9 (1.3–2.7)	0.001	1.0 (0.5–1.6)	0.842

^1^ The overall *p*-value is based on the test for heterogeneity. ^2^ Variables in **bold** were included in the final reduced model based on their overall significance at the *p* < 0.05-level. The adjusted odds ratios, confidence intervals, and *p*-vales for variables eliminated from the final model were obtained by adding each variable to the final reduced model. ^3^ Model was adjusted for 5 level school affiliation and jurisdiction. ^4^ “Ref” is the reference category used in the univariate and multivariate models. ^5^ Continuous measurements were categorised using tertiles ^6^ High category is postcodes that have high socioeconomic advantage.

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
