# Peer review of "School-Level Variation in Coverage of Co-Administered dTpa and HPV Dose 1 in Three Australian States"

_vaccines, 2021, doi:10.3390/vaccines9101202_

Round 1

Reviewer 1 Report

  1. School figures provided in Table 1 does not always add up. For e.g., “Remoteness classification” adds up to 1281, while Co-educational status” adds up to 1256, but results (line 233) says the figure as 1280. Please explain the discrepancies.
  2. Please verify if all other columns/rows add-up and if something is missing, use appropriate column/row such as “unverified” or “unavailable” or “miscellaneous”, as applicable.
  3. Use “vaccine coverage” after HPV in figure and table titles as applicable. Otherwise it gives a feeling as if HPV was tested and reported.
  4. Replace “enrollment s” in a couple of places with “enrollments”.
  5. If I understand figure 1 correctly, more schools (173) showed lower dTpa acceptance (< - 10%, a part of 32% lower dTap coverage) than those showed ‘hesitancy’ to HPV vaccine (132 with 10%+). Do authors have any reason why dTap acceptance is so low in these specific schools? Please comment/discuss.
  6. In table 2 please explain what is meant by “Ref”.

Reviewer 2 Report

I would like to commend the authors on highlighting the vaccination aspects in Australia.

As the authors stated, the few limitations in their are crucial in understanding the what & why questions, on a more granularity level.

I recommend the authors to mention certain aspects of the lines 375 - 377 as part of the conclusion in the abstract.

Throughout the manuscript, the authors have stated "more than 5% lower", which can lead to misinterpretation by the reader. - Please re-write 

Line76: "... latter 2 also provide protection against genital warts. - is this in reference to what is mentioned in like 70 - "....90% of all cases of genital warts are caused by non-oncogenic HPV types 6 & 11." If so, please state the same, else clarify.

Line98: "....... other practical barriers." - please state a few barriers as examples

Lines 288-290: "...where HPV coverage was much lower than dTpa, the HPV coverage was much lower, than in schools where dTpa was lower," - please explain as I did not understand the author's description.

At several instances in the manuscript, the authors have stated low uptake/low initiation coverage to be a result of vaccine hesitancy. I do not agree to this statement as there may be correlation b/w the 2, but may not the only causation. - this will lead to a dangerous precedent for future such studies and can lead to misinterpretation.

Lastly, the manuscript is heavily focused on data, yet there are only 2 figures, mainly line graphs. I recommend the authors to incorporate all the valuable data captured and represented in the manuscript in various graphical forms -  dot plots, scatter plots, alluvial plots, to name a few.
